# A Sensor-Based Perspective in Early-Stage Parkinson’s Disease: Current State and the Need for Machine Learning Processes

**DOI:** 10.3390/s22020409

**Published:** 2022-01-06

**Authors:** Marios G. Krokidis, Georgios N. Dimitrakopoulos, Aristidis G. Vrahatis, Christos Tzouvelekis, Dimitrios Drakoulis, Foteini Papavassileiou, Themis P. Exarchos, Panayiotis Vlamos

**Affiliations:** 1Bioinformatics and Human Electrophysiology Laboratory, Department of Informatics, Ionian University, 49100 Corfu, Greece; mkrokidis@ionio.gr (M.G.K.); aris.vrahatis@gmail.com (A.G.V.); christzouvelekis@gmail.com (C.T.); exarchos@ionio.gr (T.P.E.); 2Telesto Technologies, 15561 Athens, Greece; dimitris@telesto.gr (D.D.); fotini@telesto.gr (F.P.)

**Keywords:** biosensors, Parkinson’s disease, wearable devices, machine learning

## Abstract

Parkinson’s disease (PD) is a progressive neurodegenerative disorder associated with dysfunction of dopaminergic neurons in the brain, lack of dopamine and the formation of abnormal Lewy body protein particles. PD is an idiopathic disease of the nervous system, characterized by motor and nonmotor manifestations without a discrete onset of symptoms until a substantial loss of neurons has already occurred, enabling early diagnosis very challenging. Sensor-based platforms have gained much attention in clinical practice screening various biological signals simultaneously and allowing researchers to quickly receive a huge number of biomarkers for diagnostic and prognostic purposes. The integration of machine learning into medical systems provides the potential for optimization of data collection, disease prediction through classification of symptoms and can strongly support data-driven clinical decisions. This work attempts to examine some of the facts and current situation of sensor-based approaches in PD diagnosis and discusses ensemble techniques using sensor-based data for developing machine learning models for personalized risk prediction. Additionally, a biosensing platform combined with clinical data processing and appropriate software is proposed in order to implement a complete diagnostic system for PD monitoring.

## 1. Introduction 

The pathophysiology of Parkinson’s disease (PD) is characterized by gradual and severe degeneration of the dopaminergic neurons of the substantia nigra. The accumulation of certain symptoms in specific clusters (expressed by different phenotypic characteristics) may reflect degeneration in different neural pathways, a pathophysiological change representing different expression and progression of the disease [1]. Most cases of the disorder are likely to be caused by a complex interaction of environmental and genetic factors. Many of these cases are classified as sporadic or idiopathic due to a lack of family history. The underlying causes of such cases remain unknown. About 15% of patients with PD have a family history of this disorder, which is caused by mutations in the *LRRK2*, *SNCA*, *PARK2*, *PARK7* or *PINK1* genes [2]. Mutations in *LRRK2* are more common in familial PD with late onset. Many of the autosomal dominant gene mutations involve mitochondrial dysfunction and accumulated or dysfunctional forms of gene products. Recently, mutations in the transmembrane protein 230 (*TREM230*) gene have been linked to the familial form of PD [3]. Mutations in the DJ-1 gene (*PARK7*) are associated with familial forms of PD and therefore several studies have sought to measure DJ-1 protein in CNS and plasma samples [4]. 

To date, no specific protocol has been developed that can lead to an accurate diagnosis of the disease before the onset of motor symptoms. The clinical examination by a specialist neurologist along with the precise mapping of the patient’s history and its response to medication is the proposed combination that ultimately leads to the clinical decision that a patient suffers from the syndrome. New technologies such as electronic-Health (eHealth) systems are undergoing a real technological revolution using sensing technologies to monitor the biophysical parameters, radically changing how medical services are provided [5]. As part of this effort, accelerometers and other inertial sensors that are integrated into wearable and portable devices can be used for noninvasive vital signs monitoring of patients to detect motor dysfunctions associated with PD. These approaches succeed at a satisfactory rate to detect, in some specific movements, characteristics that identify an unusual tremor during certain activities [6]. Technological development in wearable devices has offered to the scientific community a colossal pool of sensor data for analysis and knowledge mining. Such data are usually characterized by high volume due to their high frequency of measurements and high dimensionality since multiple forms of data are traced and transferred by several various sensors simultaneously [7]. Machine learning constitutes one of the first choices to decipher such data identifying hidden patterns for their deeper and more robust analysis. Such algorithms not only can deal with the limitations of these data but also their performance is powered by large-scaled data, for example, the supervised learning algorithms. Towards this direction, there is remarkable progress on machine learning approaches for sensor-based PD data. However, as technologies and available data grow in volume and dimensionality, we need more sensor-based frameworks using machine learning processes.

## 2. Early Detection and the Need for Sensor-Based Approaches 

Molecular diagnostics comprises a powerful tool of biomedical research providing appropriate methodologies for the accurate detection of important biomarkers in various diseases. Unfortunately, the monitoring progression of neurodegenerative diseases such as PD is mainly based on clinical findings, which appear upon the extensive loss of dopaminergic neurons of the substantia nigra; therefore, the clinical diagnosis is often achieved when the disease has already progressed [8]. However, early detection of PD through understanding the pathophysiology of the diseases is vital, as it could give the patient the opportunity for an early treatment protocol, which can be valuable in eliminating further disease progression. A feature that makes the discovery of a biomarker an even more complex process is definitely the clinical heterogeneity of PD from the early preclinical to the more advanced stages of the disease. Diagnostic markers may be useful prior to the onset of motor characteristics as well as when symptoms are still insufficient to define the disease (precursor phase) or even to detect an asymptomatic population at risk of developing the disease (preclinical phase). This process could also help differentiate PD from other Parkinson’s syndromes, as misdiagnosis often occurs early in the disease and diagnostic confirmation requires autopsy [9]. The Unified Parkinson’s Disease Rating Scale (UPDRS) is currently among the most common and reliable protocols used to monitor the severity of the disease, although it remains a subjective and semiquantitative measure of motor symptoms [10]. The UPDRS questionnaire consists of four different parts, the first two of which consider active and nonmotor activities that occur in one’s daily life while the last two are completed by the doctor after examination on specific exercises that control motor functions. The third part of the questionnaire, which will henceforth be referred to as UPDRS III or motor UPDRS, is what is considered to be the basic metric to determine the stage of the patient’s disease in relation to the motor symptoms that appear. Imaging has influenced the differential diagnosis in PD in clinical practice, while at the same time genetic biomarkers contribute to the recognition of certain familial forms of the disease [11]. Therefore, there is a great need to increase the existing biomarkers available for PD and to accelerate their detection and validation at distinct stages of the disease, including molecules which can determine the patient’s transition from pre-motor to motor symptoms. Biochemical biomarkers remain also in development and can provide diagnostic utility. A-Synuclein is known to form agglomerates and is found in Lewy bodies in PD patients and thus is an important candidate marker that can be detected in cerebrospinal fluid (CSF) and blood [12]. Various growth factors have also been investigated as biomarkers of PD both diagnostically and prognostically. Low serum Brain Derived Neurotrophic Factor (BDNF) levels are associated with dopaminergic neuron loss, the severity of PD motor symptoms and cognitive symptoms [13]. Preclinical models suggest that insulin-like growth factor-1 (IGF-1) is a neuroprotective of dopaminergic neurons. Serum IGF-1 levels are elevated in patients with PD and are associated with motor function as determined by the UPDRS scale [14]. Due to the complexity of the disease, combining more than one individual biomarker from the different categories is probably the most efficient strategy for accurately predicting and diagnosing the condition and progression of the disease, including recording clinical symptoms (e.g., immunity, REM sleep behavioral disorder), genetic mutations in key genes that translate regulatory proteins (*SNCA*, *LRRK2*), analysis of biochemical parameters in biological samples (blood, CSF) and imaging techniques (PET, MRI, SPECT) [15,16]. Digital biomarkers, on the other hand, consist a valuable pool of approaches which use appropriate biosensors and tools for data collection that are converted by algorithms into a measurable result [17].

Rapid development of sensor-based systems has shown a significant increase in recent decades (Table 1). A biosensor is an analytical tool that converts a biological response into a quantifiable and editable signal [18]. The required parts comprising a typical biosensor are a bioreceptor responsible for selective identification of the target being analyzed, an interface architecture where a specific biological response takes place and generates the signal and a signal converter which translates the biorecognition event, converts it to digital and is then amplified by a circuit and sent for processing. Significant progress has been made in the development of multiple biosensor platforms for quantitative analysis and parameterization of biological signals. Simultaneous noninvasive detection of multiple analyzers is extremely attractive but at the same time requires an accurate monitoring system [19]. Additionally, the signal transfer is carried out with appropriate devices taking advantage of the ability to monitor the target analyzers on suitable biological substrates. Colorimetric analysis eliminates the need to power the sensor platform, which can facilitate the development of small and easily portable devices, while requiring additional data readers and data miners to perform sensitive measurements [20]. Real-time visual monitoring of biological indicators can be achieved using a colorimetric detection system integrated in a suitable microfluid while advanced devices allow advanced sampling and measurement based on a suitable closed system capable of collecting the sample immediately and quickly without sampling. 

Sensors for neurochemical monitoring have been fabricated such as single-walled carbon nanotubes or graphite electrodes for dopamine and uric acid determination through entrapment of nanomaterials on the surface protein-coated magnetic particles [21,22]. Gold nanoparticles and platinum and copper oxide particles have also been applied in probes for monitoring of the electrocatalytic oxidation of dopamine [23,24]. Detection platforms with continuous minimally invasive electrochemical monitoring of levodopa (L-Dopa) have gained significant progress [25]. This multimodal microfiber detection platform is based on parallel simultaneous independent enzymatic ammetric and nonenzymatic voltammetric detection of the active substance using different microneedles in the same series of spot sensors. Efforts to improve are also focused on the full integration of the wireless electronic interface, wireless transmission and the investigation of effective protective coatings while requiring clinical trials and validation in patients with PD. In this way, the researchers hope to create a closed-loop system capable of containing an L-Dopa pump and a dose automation algorithm that will replace intermittent oral administration by providing individualized drug dosing and improved disease symptom management. A PD remote monitoring system was developed based on the findings from the data collection process. It is an observational case-control study that aims to collect a set of data from PD patients and healthy volunteers by recording daytime movements, using portable sensors and detecting changes in motor symptoms after receiving an appropriate dose of medication [26].

A surface plasmon resonance-based biosensor was developed for a-synuclein autoantibody detection in undiluted blood serum using a PEGylated protein interfaces [27]. Semiconductor quantum dots such as CdSe/ZnS are generally considered suitable biocompatible fluorescence biosensors for mitochondrial complex I activity monitoring, which is highly related with the loss of PD progression [28]. Furthermore, a DNA electrochemical biosensor was utilized to determine specific nucleic acid degradation products such as 8-hydroxyguanine in urine PD samples through a molecularly imprinted polymer layer fabricated on a gold electrode [29]. Development of antibody-based biosensors offers high binding affinity and increased specificity. A monoclonal antibody fragment specific for dopamine on gold nanorods was presented, which can cleave the disulfide bond linkages and links to free thiols release [30]. Moreover, biotinylated aptamers immobilization was performed using a dopamine-HRP conjugate for competitive binding [31].

Wearable biosensors and peripherals are of great interest due to their ability to provide continuous and real-time information through dynamic, noninvasive measurements by calculating specific biomarkers in biological fluids such as sweat, tears, saliva or interstitial fluid (ISF). At the same time, they are distinguished for their high specialty, speed, portability, low cost and low power requirements [20]. Recent developments focus on electrochemical and optical biosensors, including noninvasive monitoring of specific biological molecules such as metabolites and hormones. Accurate and reliable extraction of information in real time with the use of portable technologies (biosensors, peripherals) has a wide impact on everyday life. Brain–Computer Interface (BCI) systems have been developed with purpose to provide a real-time feedback based on electrophysiological signals [32]. For example, Open-BCI offers electroencephalography (EEG), electrocardiography (ECG) and electromyography (EMG) capabilities to monitor the brain, muscles and heart on a single platform. The hardware platform is open source and compatible with Arduino [33]. A wide range of commercial portable motion monitoring devices and simple sensor networks are also available including accelerometers, gyroscopes, temperature sensors and heart rate sensors being the main components of these peripherals [34]. Given the high availability of these sensors for platforms such as the Arduino and Raspberry Pi, it is possible for researchers to easily combine a wireless portable device at very low cost to control temperature, pressure and oxygen concentration or optical sensors for heart rate and posture control. A wearable system for an objective assessment of motor performance for PD diagnosis and monitoring is proposed, composed of inertial devices for motion analysis of lower and upper limbs (two SensFoot and two SensHand) and processing algorithms that can provide a useful tool for neurologists. The system was tested on 40 healthy subjects and 40 PD patients by extracting multiple kinetic parameters and achieving an excellent discrimination with all the classifiers and all the datasets [35]. A segmented double-integration algorithm was developed to estimate step length and step time from wearable inertial measurement units. Using data from wearable inertial measurement units (IMU) sensors, the approach provides reliably and accurately measurements of spatiotemporal gait parameters that are in excellent agreement with those from widely used laboratory-based systems [36]. A sensor-based system, using embedded triaxial accelerometers from consumer smartwatches and multitask classification models, was presented to assess the amplitude and constancy of resting tremor in PD. The system was based on a deep learning multitask approach combined with the data acquired from PD patients with results showing a high agreement between the amplitude and constancy measurements obtained from a smartwatch in comparison with those obtained in a clinical assessment [37]. A multimodal deep learning model for discriminating between people with PD and without PD is shown using two data modalities, acquired from vision and accelerometer sensors in a home environment to train variational autoencoder models [38]. In another study, a mobile health (mHealth) system for continuous and objective real-life measures of patients’ health and functional mobility was tested in patients with a clinical diagnosis of PD, indicating an apparently feasible and usable approach for continuous and remote monitoring of PD patients’ functional mobility and global health status [39]. Lanoni et al. investigated how to efficiently collect movement data using six flexible wearable sensors in twenty individuals with PD for detection of bradykinesia and tremor in the upper extremities by training separate random forest classifiers and comparing the symptom detection models based on them to symptoms derived from deep convolutional neural networks [40]. Wearable technology in free-living environments could definitely help people with PD offering potential advantages to accurately detect and measure clinically relevant features and may be used in cases with a degree of confidence [41].

**Table 1 sensors-22-00409-t001:** Sensor-based approaches in PD monitoring.

Type of Sensor	Targeting/Monitoring	Ref.
Single-walled carbon nanotubes fabricated by sodium dodecyl sulfate	Simultaneous electrocatalytic determination of ascorbic acid (AA), dopamine (DA) and uric acid (UA)	[21]
Nanosized copper oxide/multiwall carbon nanotubes	Electrocatalytic oxidation of dopamine monitoring	[23]
Microneedle sensing platform	Electrochemical monitoring of levodopa (enzymatic–amperometric and nonenzymatic voltammetric detection)	[25]
Electroanalytical assay using alpha-synuclein modified electrodes	a-synuclein detection through autoantibodies sampling	[27]
Semiconductor quantum dots (CdSe/ZnS)	Mitochondrial complex I activity fluorescence monitoring	[28]
DNA electrochemical biosensor through an imprinted polymer layer fabricated on a gold electrode	Nucleic acid degradation products determination (8-hydroxyguanine)	[29]
Antibody-based biosensor on multiblock nanorods (Au and Ag)/biotinylated aptamers immobilization	Dopamine detection	[30,31]
A segmented double-integration algorithm	Calculation of step length and step time from wearable inertial measurement units, spatiotemporal gait parameters measurement	[36]
Embedded triaxial accelerometers from consumer smartwatches and multitask classification models	Assessment of the amplitude and constancy of resting tremor	[37]
MCPD-Net, a multimodal deep learning model using visions accelerometer sensors	Effective representations of human movements prediction	[38]
mKinetikos, a mobile-based system (mHealth system)	Continuous and remote monitoring of PD patients’ functional mobility and global clinical status	[39]
Flexible wearable sensors attached to the hands, arms and thighs	Detection of bradykinesia and tremor in the upper extremities	[40]

## 3. The Sensor Perspective 

Herein, a framework for the design of an accurate and reliable system that can be easily applied in clinical practice to monitor the diagnosis and management of PD is presented. The objective of the proposed integrative heterogeneous tool is to make the results readily available and visible to users via a single workflow. Through the interconnection and integration of systems, the homogenization of the pathway of extracting, correlating and retrieving data can be achieved. The complete diagnostic platform consists of the following building blocks, as depicted in Figure 1: (i) a biosensor properly designed for the measurement of specific proteins in biological fluids, among of which could be α-synuclein, total and phosphorylated tau, apolipoproteins A and E, β2-microglobulin and amyloid Ab42; (ii) a microprocessor with the appropriate analog processing peripherals to amplify and convert to digital the analog signal received from the biosensor (ADC—Analog to Digital Converter); (iii) BITalino and Mysignals peripheral sensor platforms for monitoring and measuring biometric parameters of examinees (electrocardiogram, electromyogram, temperature, breathing rate, blood pressure, body scale parameters); (iv) central collection and management unit (Single Board Computer (SBC)—Raspberry Pi) that allows data management received from the connected peripheral platforms as well as from the biosensor and (v) a server for storing, managing, viewing and processing data sent by the SBC, which also hosts the decision support system and the user interface. The display of the results to the end user (medical doctor) can be performed evenly with the help of a central visual representation of processed data and data of the examinee (dashboard), without the need to understand the process of collecting and converting data from different sources of the individual systems (biosensor or peripheral sensor platforms). The collection of information can be executed by directly reading the corresponding database (dynamic queries). In addition, to achieve the interconnection of systems and databases, the appropriate Application Programming Interface (APIs) must be provided. One of the main advantages of the API is that it allows information to be exported between systems. If dynamic queries are used in the database, then an interface would be used between the programming language and the database through which they communicate. To that end, there is Open Database Connectivity (ODBC) (general application) and Java Database Connectivity (JDBC) (for Java language). ODBC is a widely used API for accessing and manipulating databases. The aim for each application is to have access to any data, regardless of the system. ODBC creates an intermediate layer (middle layer, database driver) between the application and the database, which translates the application queries into commands that the database can understand.

In this perspective, the development of a suitable polymeric substrate for the biosensor is necessary for accurate detection and quantification of the levels of the measured biomarker [42] whereas the selection of specific application-development platforms for medical devices such as MySignals platform, low-cost biosignal acquisition systems such as BITalino platform together with a central unit for collecting and managing measurements from the biosensor and the peripheral sensors of the platform plays a pivotal role. Specific criteria should also be taken into account during the developmental process, such as rigorous specifications and operation of a diagnostic device, the intermediate software along with the interface completion, in which the interconnection of the diagnostic device with all the sensors (peripheral and biosensor), along with the management units (Raspberry Pi and server) [43]. Given the capabilities provided by the peripheral platforms for interfacing with other systems through API programming, the integration with the software applications and systems defined for the offered solution is easily achieved. Some available options of APIs on offer include device management, functionality and call answering. For example, the data in the proposed layout can be performed using the “OpenSignals (r)evolution” software for the BITalino platform and each record can be stored via the software framework in a text file, which will then be processed through a developed code (e.g., in Python). Mysignals platform has a similar API. 

Focusing on the design of the biochemical sensor as a critical part of the diagnostic device, this stage requires a detailed development process to accomplish the required functional milestones such as accuracy, linearity, selectivity and specific interaction with the target. Signal detection and modification is basically electrochemical and based on the measurement of the total conductivity of the system [44]. It detects alterations in the number of ions, their charge and their mobility that depend on the biological response. The glass slides are coated with the polymer substrate polyaniline and the surface is modified accordingly to attach the selected proteins for bioidentification [45]. The activation step at a certain temperature by the method of silanization is considered an important factor for the formation of activation bonds on the surface of the glass. The polymerization of the monomer imparts good mechanical properties, stability and conductivity to the system. The operation principle of the proposed biochemical monitoring sensor is based on the assumption that the interactions between polymer and protein can change the density of the conductivity carriers. Conducting polymers have the potential to be used for biosensor applications and the proposed template can act as a biomolecular sensor. Similar polymer series and sensing devices have been developed, which incorporate a film of polymer acid-doped polyaniline to respond to proteins at physiological pH and ionic strength [46,47]. Recently, a capacitive field-effect sensor with a nanomaterial-modified surface supported by a polyelectrolyte polyallylamine hydrochloride layer for bioreceptor coupling was presented [48]. It should be noted that the emeraldine salt of polyaniline was selected in our system, which presents a characterized conductivity alteration in response to pH as previously identified [49]. 

In terms of peripheral sensor platforms, BITalino system has been exhibited as an equivalent recording device for stationary ECG recordings in psychophysiological experiments [50]. Furthermore, Ponciano et al. used this wearable device to identify the mobility’s initiation [51], whereas the ECG and Galvanic Skin Response (GSR) signals were collected with the BITalino (r)evolution kit platform trying to unravel the impact of using personality assessments [52]. On the other hand, Mysignals was used to develop eHealth embedded systems for monitoring elderly health based on the Internet of Things (IoT) and Fog computing [53] for calculating specific biometric markers related to predictive analysis in the field of eldercare, such as the connection between noncommunicable diseases and cognitive impairments [54], and to generate vital signs and data for medical patients [55]. Lastly, many studies adopt Raspberry Pi, a low-cost computing platform that is profitably applied in the field of IoT and embedded systems [56].

Herein, the BITalino R–IoT module can be proposed, which is based on the CC3200 SimpleLink Wi-Fi chip class that incorporates the ARM^®^ Cortex^®^-M4 Core microcontroller at 80 MHz in the BITalino R–IoT Barebone version. The CC3200 is a combination of two ARM processors, one that controls the WiFi modem and the network stack and a second one, which is the application processor that executes the user password. Moreover, it can communicate via Bluetooth to collect and process data and can send it to the digital-to-analog converter without delay. The Cortex-M4 processor, through the firmware, can support the Thread or Handler modes, and can be run in Thumb or Debug modes. Additionally, the processor may restrict or block access to certain resources by executing code in privileged or unprivileged mode while it is intended for high performance applications and embedded applications that require fast shutdown features. The main reason for using the ARM processor is the reduction of the total space required in memory. The Thumb2 command line is an upgrade of an existing 16-bit Thumb and allows 32-bit commands to be converted from 16-bit commands to a single program. The additional 32-bit commands used by Thumb2 cover all the functionality required by ARM. Plugged version can also be used where the sensors are not integrated in the board, but they can be connected with an RJ-22 cable and a Molex Sherlock terminal (on the sensor side). This configuration offers more flexibility and adaptability.

On the other hand, MySignals development board is based on the Libelium IoT Core architecture with an 8-bit ATmega 2560 microcontroller. The high-performance, low-power 8-bit AVR microcontroller is based on RISC architecture. It combines a 256 KB ISP flash memory, a 8 KB SRAM, an 4 KB EEPROM, a real-time meter, 6 flexible meters, PWM, 4 UARRTs and a 10-bit A/D transformer with 16 channels. It has an output of 16 MIPS at 16 MHz and its operating voltage is 4.5–5.5 V. Performing multiple commands in one clock cycle can achieve a performance of 1 MIPS per MHz by balancing power consumption and processing speed. Depending on the challenges that arise and to overcome potential challenges, the idle mode, the ADC Noise reduction mode, the power-down mode, the power-save mode or the standby mode can be chosen.

Lastly, we could summarize the proposed architectural representation as a clear discrimination of the systems into three levels: the client level, the middleware level and the diagnostic data base level. The role of middleware for collecting and forwarding peripheral sensor data is performed by BITalino/MySignals software with Arm Cortex M4/Atmega 2560 processor, while Raspberry Pi software wirelessly collects (Bluetooth protocol) the signals of the various sensors from the two peripheral platforms, processes them by forwarding them through the server (TCP/IP protocol) to the diagnostic database in the appropriate format, recognizable by the database. The database, running queries and applying the built-in identification algorithms, taking into account the installed knowledge base and through the conclusion mechanism, leads to the decision on the sensor finding. Then, the decision is forwarded through the server to the main portal/dashboard for evaluation by the user (medical doctor).

Among the main limitations of the proposed framework during the development process, note that the possible stereochemical interference leading to analyte inactivation due to high density or denaturation process resulting from potent interactions with the biochemical sensor’s surface, the sequential connection of contacts of the polymer film, the appropriate measuring board through the ADC software, together with storage of the measurements in the central management unit of the Raspberry Pi device. Under these circumstances, the continuous control for communication between the different parts of the framework should be performed for receiving accurate measurements from the biochemical and the peripheral sensors, the central Raspberry Pi signal collection and the processing unit. Moreover, a signal normalization process method is strongly recommended. Best practices for clinical data collection should ensure that stakeholders have all the relevant guidelines required both to understand their obligations and to submit the necessary information. Appropriate installation of systems for data collection, organization and recording is highly required, including the automation of the medical record, the provision of decision support, the extraction of comparative results and the provision and utilization of clinical guidelines. Apart from MySignals and BITalino, Shimmer can provide an eHealth platform for the measurement of several signals such as ECG, EEG and EMG. Clinical data of PD patients are mainly characterized by complexity in order to capture them in an information system. Detailed analytical processes including targeted design and complex structures are necessary. Data integration platforms and systems through collection, storage and transmission of clinical information are widespread, such as p-BioSPRE, which provides secure data sharing for personalized treatment, and STRIDE with its applications in translation research that provides standardized information [57,58]. 

## 4. Recent Machine Learning Advancements in Sensor-Based Data

When aiming to gain a clinical understanding of wearable PD data, complex statistical machine learning algorithms are usually required. Supposing that wearables are proven practically valuable for clinical applications, many complications might obfuscate the process and should be avoided. Kubota et al.’s survey analyzes many of these issues and examines how they can be managed [7]. The authors pose a question regarding achieving the full potential of wearable sensor data in PD with the help of current technological advances in this field. A set of significant steps can lead the way, such as data sharing, clinical validation and open platform standardization and dissemination [59]. Healthcare applications use wearable sensors and provide a large volume of time-series data that present significant challenges in analysis. Analyzing gait disturbances in PD can be a very challenging procedure as these sensors can produce time-stamped datasets.

Noninvasive approaches may have potential value as diagnostic support tools and people can be monitored passively in their homes, which was indicated by a pilot study measuring PD symptoms from 10 individuals with PD and 10 controls via a smartphone and a remote visit by a specialist per week [60]. The validity of smartphone-based sensor technologies in clinical trial setting and the smartphone-derived severity score for PD can provide an adequate measure of motor symptoms, which was shown though a phase 1 PD clinical trail including 44 PD participants and 35 healthy controls [61], as well as using smartphone sensor data from PD individuals and a novel machine learning approach [62]. Moreover, the Parkinson@Home validation study provided a new reference dataset for the development of digital biomarkers to monitor persons with PD in daily life [63], whereas wearable inertial systems such as a wearable sensor-based arm swing algorithm for patients with PD has been developed and validated presenting high accuracy and showing great potential to be used in a daily-living environment [64].

From the computational and mathematical perspective, the sensor-based biomedical data are quite complex due to their high volume or/and their dimensionality. Even though most of the PD wearable data seem to have high dimensionality because of an excessive number of different measurements, there is a possibility that the intrinsic dimensionality of the data is conspicuously small. Machine learning methods have gained ground in recent years regarding the processing of wearable PD data. Artificial neural networks are the first choice to manage this complexity, thus several approaches have been published recently. Indicatively, the work in [65] proposes an application of a comprehensive empirical assessment of Convolutional Neural Networks (CNNs) on a large-scale image classification of gait signals. Its main goal was to assist clinicians and the general public in achieving early disease diagnoses. According to the experimental findings, the suggested models exceeded the performance of the existing state-of-the-art similar tools concerning the degree of correctness. Deep learning approaches have shown remarkable results in early detection of Parkinson’s disease [66]. For example, Jane et al. aimed to develop an efficient clinical decision-making system (CDMS) that facilitates the ability for doctors to diagnose the severity of gait disturbances of patients who suffer from PD [67]. Thus, the authors demonstrate a classifier that creates a temporal classification model for CDMS prediction and classification, called Q-Backpropagated Time-Delay Neural Network (Q-BTDNN). 

While deep learning approaches have overtaken the scientific community in recent years in the machine learning field, their inherent complexity is often characterized as black box, which makes it difficult to interpret the results. To avoid this issue, approaches under the perspective of ensemble learning relying on tree-based methods can be used, which have shown promising results in PD studies. More specifically, the work of Hughes et al. was motivated by the necessity of securing the efficacy of diagnosis and treatments using a computational framework that monitors the patient’s gait [68]. In this study, the Extreme Gradient Boosting (XGBoost) and Artificial Neural Network (ANN) models were applied. XGBoost belongs to the ensemble classification category, which aims to create a robust classifier by integrating the decisions from multiple individual classifiers. Al-Sarem et al. proposed several set techniques for detecting PD, including random forest, XGBoost and categorical boosting (CatBoost) [69]. They exploit the inherent feature of these tree-based ensemble methods, which is to investigate the features (dimensions) that are dominant in the case under study. A better interpretation can be achieved in various prediction tasks for PD by identifying those features where a classifier can separate the classes in a dataset. Almeida et al. attempted to use various feature extraction and machine learning strategies to detect PD. In their research, the authors analyzed Multilayer Perceptron (MLP), Support Vector Machines (SVM), Optimum Path Forest and K-Nearest Neighbors (K-NN) according to their ability for effective classification. As a result, they found that the most efficient activity for detecting PD is phonation [70]. Nahar et al. suggest a type of detection of PD that relies on machine learning and uses feature selection and classification techniques [71]. The feature selection procedure uses Recursive Feature Elimination (RFE), Boruta and Random Forest (RF) classifier. Bagging, gradient boosting, XGBoost and Extra Tree Classifier are four classification algorithms that were used to detect PD. Finally, the technique that outperformed the other three was bagging with RFE.

Recent literature has a plethora of these approaches. However, the biological complexity of PD and the continuous increase of sensor-based PD data create the impression that this field is in its infancy. Such data open new challenges, and machine learning has the potential to provide answers. Part of this challenge is provided by the ensemble supervising learning techniques that, through hybrid models, can give promising results. Details for an indicative workflow are given in the following section.

## 5. Ensemble Methods in Sensor-Based Data—Towards the Future Big Challenge

Ensemble methods combine multiple classification models for better predictive performance compared to applying a single classifier. The models differ in the type of classifier or the search space, increasing the diversity of a case under study. The rationale behind using this strategy is that one specific algorithm in a particular dataset may not extract reliable results for various reasons, such as when the algorithm does not fit or overfits this dataset. We also need high diversity when dealing with data with noise, high complexity and non-separable classes. Therefore, the classification process using multiple classifiers or multiple subspaces can cover part of this complexity.

There are three main ensemble classification categories: stacking, bagging and boosting methods [72]. The stacking methods apply different classifiers, which are applied in parallel, creating a new classification model based on the predictions of the previous ones. Usually, for a dataset, different classification algorithms are applied, and a voting scheme determines each sample’s class. The intuition here is that each algorithm highlights different data aspects, and if the majority is driven to a specific direction, this may be the correct one.

The bagging method combines many classification processes, where each classifier uses a bootstrap sample of the original training set. The same algorithm is applied to different datasets, which are subspaces of the original space. Similarly, a voting scheme aggregates and defines the predicted class of a given sample. The various subspaces from the original space increase the data diversity, offering more accurate results in complex data with noise or data with dominant features. A typical example of a bagging algorithm is the Random Forest classifier.

Boosting methods try to construct a strong classifier by integrating multiple weak classifiers and focusing on their errors. Its training phase is an iterative process where each model tries to correct the errors from the previous model until a given threshold related to the train error of the number of added models is reached. An indicative traditional boosting method is the AdaBoost algorithm, which boosts the performance of decision trees on binary classification tasks. 

In the ensemble techniques, we can also incorporate dimensionality reduction algorithms, i.e., to apply several classification models on transformed data spaces (from the original space) through dimensionality reduction methods (indicative workflow in Figure 2). In this case, the nondeterministic dimensionality reduction algorithms operate better, since we obtain different transformed data spaces through their stochastic processes [73]. Thus, we achieve high diversity [74], a key point for a better classification process in a complex dataset with noise and non-separable categories. The main advantage here is utilizing lower-dimensional space in the feature space, avoiding limitations such as managing the curse of dimensionality. A crucial point is that the dimensions are reduced without losing the initial data structure by keeping their pairwise distances within a given allowable error [75]. Indicative methodologies under this perspective are the random projection method, the autoencoders and the stochastic neighbor embeddings.

We conclude that the increase in sensor-based data enhances the complexity we have to manage, in terms of volume as in dimensionality. Dimensionality reduction methods can address the high data dimensionality, while the ensemble methods have provided evidence for their reliable results in supervised learning processes. Hence, the combination of these two techniques can create a hybrid model able to manage the sensor-based data challenges. Their integration offers a search space with high diversity, with reduced dimensions where a classifier can operate better, a voting system that can determine the predominant sample’s class, and many other advantages. Additionally, the ensemble methods are traditionally slower than single classifiers. However, searching in a diminished space can cover this weakness. The plethora of both ensemble and reduced dimensions can offer many reliable tools for sensor-based data. We estimate the developments in this direction can clarify the complex process posed by PD.

## Figures and Tables

**Figure 1 sensors-22-00409-f001:**
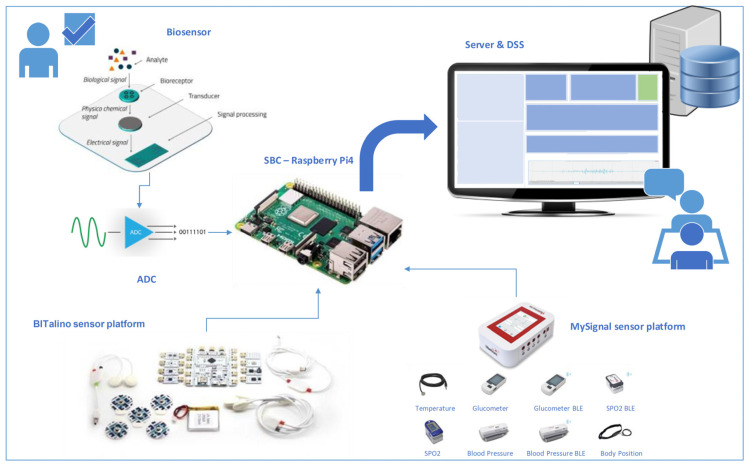
Integrated diagnostic layout for Parkinson’s disease monitoring.

**Figure 2 sensors-22-00409-f002:**
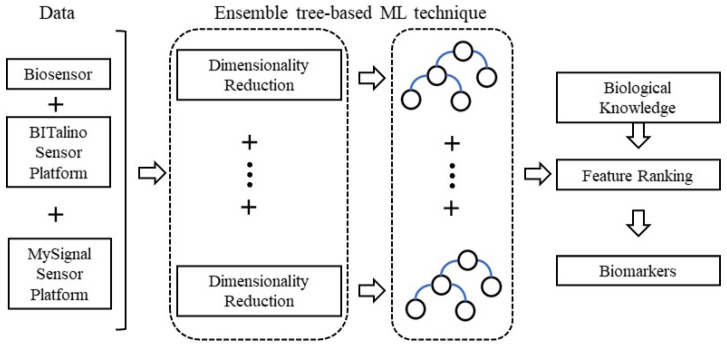
A proposed framework for early-stage PD detection using ensemble learning with dimensionality reduction methods.

## Data Availability

Not applicable.

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
