# Peer review of "A Sensor-Based Perspective in Early-Stage Parkinson’s Disease: Current State and the Need for Machine Learning Processes"

_sensors, 2022, doi:10.3390/s22020409_

Round 1
Reviewer 1 Report
This manuscript presents a position paper with literature review of some of the on going work in the area of sensor-based monitoring of Parkinson’s disease (PD). The literature review is not systematic, which is understandable given the vast amount of work under this fairly generic topic. The paper is organized around: reviewing digital biomarkers derived from neurochemical monitoring sensors; use of IMUs for free-living monitoring of PD patients; a proposed framework for end-to-end monitoring infrastructure; and a review of some of the different machine learning methodologies proposed for remote monitoring of PD.
The review aspect of the manuscript is quite extensive, although I will mention few gaps worth noting below. My main concern is with the organization of the paper. For the most parts, it is presented as a position paper which points to some key literature, but then in Section 2 the authors propose a framework for building a monitoring tool. The proposed framework is sound, but it does not become clear: what aspects of this system have actually been tested in studies; how does this approach compare to other studies; are there different ways of tackling the data collection; what are the hardware risks/limitations of the proposed approach.
Section 3 of the manuscript then moves on to discussing some algorithmic challenges in PD digital monitoring, but it seems to be confusingly organized on the basis of algorithms used in the respective studies, rather than the clinical task addressed in each work. There are huge underlying differences: behind in-lab detection of PD vs free-living detection of PD; active digital monitoring of PD vs passive monitoring [1]; monitoring of PD severity vs binary detection [2]; prodromal detection of PD vs late onset [3]; digital monitoring of gait, tremor, voice, dexterity or others [4]; quality control of free-living data. The complexity and merit of the mathematically different algorithms involved surely would depend on a deeper discussion about the tasks for which they were deployed. To me the distinction behind resources reviewed in the Introduction and here is not made clear. While I agree with the general conclusion that this field has much room to grow, it would be beneficial for the readers to be pointed to some of the larger initiatives looking at potential of digital technologies in clinical trials [5][6].
The manuscripts concludes with a summary of ensemble methods and discussion on their appropriateness for various digital monitoring tasks. This section includes surprisingly few actual references demonstrating the previous use of methods such as random forests in digital monitoring of PD. Furthermore, a lot of the challenges related to digital monitoring of PD are not simply predictive, so the manuscript should discuss some of the downsides of ensemble methods such as their black-box nature. More and more work has focused on doing correct causal inference in digital monitoring which goes against simply picking the method performing best against some empirical measure of accuracy.
The manuscript could benefit from including resource on actual unconstrained free-living monitoring of PD with wearables.
Minor issues:
Line 22: “This work examines the facts and current situation of sensor-based approaches in PD diagnosis” - as authors point out themselves later in the manuscript, digital monitoring is in early stages, so hoping to summarize all the “facts” in the field is overly ambitious task.
Figure 1: resolution of the figure does not allow to read some of the smaller black font text
Just some examples of large scale deployment include [4] where active smartphone tests where deployed in patients homes; [1], [5] and [6] where people were monitored passively in their homes. [8] which studied possibility of capturing reduced arm swing common in PD and [8] showed that certain gait digital biomarkers can be predictive of prodromal PD.
Authors discuss the challenges of using established PD measures such as the UPDRS. There has been recent work gaining traction, on using different scores designing for digital measurement of PD [9].
References:
[1] Evers, L.J., Raykov, Y.P., Krijthe, J.H., De Lima, A.L.S., Badawy, R., Claes, K., Heskes, T.M., Little, M.A., Meinders, M.J. and Bloem, B.R., 2020. Real-life gait performance as a digital biomarker for motor fluctuations: The Parkinson@ Home validation study. Journal of medical Internet research, 22(10), p.e19068.
[2] Lipsmeier, F., Taylor, K.I., Kilchenmann, T., Wolf, D., Scotland, A., Schjodt‐Eriksen, J., Cheng, W.Y., Fernandez‐Garcia, I., Siebourg‐Polster, J., Jin, L. and Soto, J., 2018. Evaluation of smartphone‐based testing to generate exploratory outcome measures in a phase 1 Parkinson's disease clinical trial. Movement Disorders, 33(8), pp.1287-1297.
[3] Maetzler, W. and Hausdorff, J.M., 2012. Motor signs in the prodromal phase of Parkinson's disease. Movement Disorders, 27(5), pp.627-633.
[4] Arora, S., Venkataraman, V., Zhan, A., Donohue, S., Biglan, K.M., Dorsey, E.R. and Little, M.A., 2015. Detecting and monitoring the symptoms of Parkinson's disease using smartphones: A pilot study. Parkinsonism & related disorders, 21(6), pp.650-653.
[5] Marek, K., Jennings, D., Lasch, S., Siderowf, A., Tanner, C., Simuni, T., Coffey, C., Kieburtz, K., Flagg, E., Chowdhury, S. and Poewe, W., 2011. The Parkinson progression marker initiative (PPMI). Progress in neurobiology, 95(4), pp.629-635.
[6] Bloem, B.R., Marks, W.J., De Lima, A.S., Kuijf, M.L., Van Laar, T., Jacobs, B.P.F., Verbeek, M.M., Helmich, R.C., Van De Warrenburg, B.P., Evers, L.J.W. and van de Zande, T., 2019. The Personalized Parkinson Project: Examining disease progression through broad biomarkers in early Parkinson’s disease. BMC neurology, 19(1), pp.1-10.
[7] Warmerdam, E., Romijnders, R., Welzel, J., Hansen, C., Schmidt, G. and Maetzler, W., 2020. Quantification of arm swing during walking in healthy adults and parkinson’s disease patients: Wearable sensor-based algorithm development and validation. Sensors, 20(20), p.5963.
[8] Del Din, S., Elshehabi, M., Galna, B., Hobert, M.A., Warmerdam, E., Suenkel, U., Brockmann, K., Metzger, F., Hansen, C., Berg, D. and Rochester, L., 2019. Gait analysis with wearables predicts conversion to parkinson disease. Annals of neurology, 86(3), pp.357-367.
[9] Zhan, A., Mohan, S., Tarolli, C., Schneider, R.B., Adams, J.L., Sharma, S., Elson, M.J., Spear, K.L., Glidden, A.M., Little, M.A. and Terzis, A., 2018. Using smartphones and machine learning to quantify Parkinson disease severity: the mobile Parkinson disease score. JAMA neurology, 75(7), pp.876-880.
Author Response
"This manuscript presents a position paper with literature review of some of the on going work in the area of sensor-based monitoring of Parkinson’s disease (PD). The literature review is not systematic, which is understandable given the vast amount of work under this fairly generic topic. The paper is organized around: reviewing digital biomarkers derived from neurochemical monitoring sensors; use of IMUs for free-living monitoring of PD patients; a proposed framework for end-to-end monitoring infrastructure; and a review of some of the different machine learning methodologies proposed for remote monitoring of PD.
The review aspect of the manuscript is quite extensive, although I will mention few gaps worth noting below. My main concern is with the organization of the paper. For the most parts, it is presented as a position paper which points to some key literature, but then in Section 2 the authors propose a framework for building a monitoring tool. The proposed framework is sound, but it does not become clear: what aspects of this system have actually been tested in studies; how does this approach compare to other studies; are there different ways of tackling the data collection; what are the hardware risks/limitations of the proposed approach."
Response: Thank you for your kind commentary and your critical points. Regarding the organization, we agree with your opinion as a position paper so we changed the type of the article as a perspective. In order to strength our framework, we added few studies which also follow the different parts of our proposed system and highlight the utility of the combination of more than one individual building block from the different categories. We clearly stress the biochemical utility provided through the suggested neurochemical sensor characterized by a specific polymer-modified surface as well as the extensive use of accelerometers and other wearable and portable devices for non-invasive vital signs monitoring in clinical practice. Furthermore, discussion about data collection and the advantages and limitations of our integrated layout is also included, however our system comprises a proposed approach and decisively clinical symptoms recording is necessary for accurate evaluation.
"Section 3 of the manuscript then moves on to discussing some algorithmic challenges in PD digital monitoring, but it seems to be confusingly organized on the basis of algorithms used in the respective studies, rather than the clinical task addressed in each work. There are huge underlying differences: behind in-lab detection of PD vs free-living detection of PD; active digital monitoring of PD vs passive monitoring [1]; monitoring of PD severity vs binary detection [2]; prodromal detection of PD vs late onset [3]; digital monitoring of gait, tremor, voice, dexterity or others [4]; quality control of free-living data. The complexity and merit of the mathematically different algorithms involved surely would depend on a deeper discussion about the tasks for which they were deployed. To me the distinction behind resources reviewed in the Introduction and here is not made clear. While I agree with the general conclusion that this field has much room to grow, it would be beneficial for the readers to be pointed to some of the larger initiatives looking at potential of digital technologies in clinical trials [5][6]."
Response: We thank the reviewer for this comment, that certainly increased the quality of the manuscript. Our work aims to show a sensor-based perspective in early-stage Parkinson’s disease as well to highlight the state-of-the-art in the Machine Learning field concerning the analysis of such data. For this reason, we change our status from “Review paper” to “Perspective paper”. We also highlight the suggested general ensemble learning model with dimensionality reduction techniques. We have made relevant corrections in Section 3 and added the suggested references.
"The manuscript concludes with a summary of ensemble methods and discussion on their appropriateness for various digital monitoring tasks. This section includes surprisingly few actual references demonstrating the previous use of methods such as random forests in digital monitoring of PD. Furthermore, a lot of the challenges related to digital monitoring of PD are not simply predictive, so the manuscript should discuss some of the downsides of ensemble methods such as their black-box nature. More and more work has focused on doing correct causal inference in digital monitoring which goes against simply picking the method performing best against some empirical measure of accuracy."
Response: We couldn’t agree more with the reviewer, and in fact this is the core of our paper. Neural networks are black box with poor interpretability, while our proposed workflow, which relies on ensemble tree-based methods, avoids this problem. We added this argument in Section 3.
"The manuscript could benefit from including resource on actual unconstrained free-living monitoring of PD with wearables."
Response: Thank you for your suggestion. The last paragraph of Section 2 is already fully oriented on wearable biosensors and peripherals with emphasis on PD patients. We stressed the term ‘‘free-living monitoring’’ and we highlight the impact of remote, non-invasive approaches in daily living environment (see Section 3).
"Minor issues:
Line 22: “This work examines the facts and current situation of sensor-based approaches in PD diagnosis” - as authors point out themselves later in the manuscript, digital monitoring is in early stages, so hoping to summarize all the “facts” in the field is overly ambitious task."
Response: Thank you for your suggestion. We replace ‘‘This works examines the facts’’ with ‘‘This work attempts to examine some of the facts’’.
"Figure 1: resolution of the figure does not allow to read some of the smaller black font text"
Response: We revised Figure 1 accordingly.
"Just some examples of large scale deployment include [4] where active smartphone tests where deployed in patients homes; [1], [5] and [6] where people were monitored passively in their homes. [8] which studied possibility of capturing reduced arm swing common in PD and [8] showed that certain gait digital biomarkers can be predictive of prodromal PD. Authors discuss the challenges of using established PD measures such as the UPDRS. There has been recent work gaining traction, on using different scores designing for digital measurement of PD [9]."
Response: Thank you for your comment. We have expanded the manuscript and included your suggestions in the revised draft.
Reviewer 2 Report
Title: A sensor-based perspective in early-stage Parkinson’s disease: 2 Current state and the need for Machine Learning processes
Summary: The review paper analyzes the current state of the art in the analysis of Parkinson's disease and the need for machine learning algorithms for better diagnosis of PD
Comments:
- As written, it is very hard to follow the review. It feels more like a linear description of various papers rather than a methodological survey/review. It would be better if the authors clearly identify what are the focus areas and present tables/figures to show how the field has progressed.
- The relevance of the hardware platform to the review is not clear. Why to focus on these particular devices? A more broad analysis of the types of devices used is probably useful.
- In general, more visualizations and tables will help the readability of the paper.
- Please define what is NP in line 40 and other places.
Author Response
"Title: A sensor-based perspective in early-stage Parkinson’s disease: 2 Current state and the need for Machine Learning processes
Summary: The review paper analyzes the current state of the art in the analysis of Parkinson's disease and the need for machine learning algorithms for better diagnosis of PD
Comments:
- As written, it is very hard to follow the review. It feels more like a linear description of various papers rather than a methodological survey/review. It would be better if the authors clearly identify what are the focus areas and present tables/figures to show how the field has progressed."
Response: Thank you for your comments. This work attempts to examine some of the facts and current situation of sensor-based approaches in PD diagnosis (neurochemical and wearable monitoring sensors) and discusses ensemble techniques using sensors-based data for developing machine learning models for personalized risk prediction. On this direction, we propose a biosensing platform consists of different building blocks and combined with clinical data processing and appropriate software in order to implement a complete diagnostic system for PD monitoring. To better reflect the content, we changed the type of the article from review to perspective. Moreover, we have proceeded with a lot of changes and additions for clarification of the content.
"- The relevance of the hardware platform to the review is not clear. Why to focus on these particular devices? A more broad analysis of the types of devices used is probably useful."
Response: Thank you for your comment. We stressed more the selection of the type of devices and we included further studies which can support our suggested framework. The integrated system of both biochemical sensors (for PD fluid biomarkers measurement) along with well-established Bitalino and MySignal wearable platforms, collection and management unit and server for storing and processing data may support PD symptoms monitoring and shed light on the complex process posed by the disease.
"- In general, more visualizations and tables will help the readability of the paper."
Response: A new Figure 2 was added showing a proposed framework for early-stage PD detection using ensemble learning with dimensionality reduction methods.
"- Please define what is NP in line 40 and other places."
Response: We replaced NP (PD-related neurodegenerative phenotype) with PD.
Reviewer 3 Report
The structure of the paper is confusing. It is difficult to tell if it is a review paper, a systematic or a perspective paper. If it aimed to be a review paper, it fails the methodology, and if was to be a perspective paper, the authors explore a wide field of research such as biosensors for biomarkers to sensors to monitor motion. Therefore, the paper cannot be published under its current form. I strongly recommend to either restructure it with a specific type of sensor ( biosensor or motion sensor) and an explicit description of the advantages and limitations of the various machine learning methods or to describe the methodology if they prefer it to be a review paper.Author Response
"The structure of the paper is confusing. It is difficult to tell if it is a review paper, a systematic or a perspective paper. If it aimed to be a review paper, it fails the methodology, and if was to be a perspective paper, the authors explore a wide field of research such as biosensors for biomarkers to sensors to monitor motion. Therefore, the paper cannot be published under its current form. I strongly recommend to either restructure it with a specific type of sensor (biosensor or motion sensor) and an explicit description of the advantages and limitations of the various machine learning methods or to describe the methodology if they prefer it to be a review paper."
Response: Thank you for your comments. We agree with your statement and we replaced the type of the article as perspective. In our opinion as well as following the commentary from the other Referees, we should give emphasis on biochemical and wearable sensor as our proposed framework includes both of the type of monitoring systems. Moreover, a paragraph about limitations has been added in section 2. Finally, we have extended the manuscript and added several studies which examine sensor-based approaches as well as machine learning methodology.
Round 2
Reviewer 1 Report
Thank you for addressing my comments above, with the edited submission type, I think this manuscript is suitable for publication as a Perspective paper.
Author Response
Comment:
"Thank you for addressing my comments above, with the edited submission type, I think this manuscript is suitable for publication as a Perspective paper. "
Response:
Thank you very much for your kind words and your suggestion for acceptance of our manuscript for publication.
Reviewer 2 Report
- In the revised version the authors say that a biosensor platform is proposed. However, without any experimental validation it is hard to know the validity of the platform for PD analysis and diagnosis. Therefore, the reviewer recommends the authors to do some analysis of the data collection and processing capabilities of the platform. It does not have to be with PD patients. Rather, it can be on gesture recognition, activity recognition or any other application. This validation will help in proving the effectiveness of the platform.
- Tables to summarize the papers would be useful
Author Response
Comment 1:
"In the revised version the authors say that a biosensor platform is proposed. However, without any experimental validation it is hard to know the validity of the platform for PD analysis and diagnosis. Therefore, the reviewer recommends the authors to do some analysis of the data collection and processing capabilities of the platform. It does not have to be with PD patients. Rather, it can be on gesture recognition, activity recognition or any other application. This validation will help in proving the effectiveness of the platform."
Response:
Thank you for your comment. Future work will focus on the validation of the biosensor platform, through a proof of concept study, measuring and comparing parameter results in patients with Parkinson's Disease and healthy controls.
Comment 2:
"- Tables to summarize the papers would be useful."
Response:
Thank you for your suggestion. We added a new Table which highlights characteristic sensor-based approaches in PD monitoring.
Round 3
Reviewer 2 Report
Thanks for addressing the comments.